# Synovial Membrane Is a Major Producer of Extracellular Inorganic Pyrophosphate in Response to Hypoxia

**DOI:** 10.3390/ph17060738

**Published:** 2024-06-05

**Authors:** Émilie Velot, Sylvie Sébillaud, Arnaud Bianchi

**Affiliations:** Université de Lorraine, CNRS, IMoPA, F-54000 Nancy, France; emilie.velot@univ-lorraine.fr (É.V.); sylvie.sebillaud@inrs.fr (S.S.)

**Keywords:** synovial fibroblasts, extracellular inorganic pyrophosphate (ePPi), ectonucleotide pyrophosphatase/phosphodiesterase 1 (ENPP1), Ank, hypoxia

## Abstract

Calcium pyrophosphate dehydrate (CPPD) crystals are found in the synovial fluid of patients with articular chondrocalcinosis or sometimes with osteoarthritis. In inflammatory conditions, the synovial membrane (SM) is subjected to transient hypoxia, especially during movement. CPPD formation is supported by an increase in extracellular inorganic pyrophosphate (ePPi) levels, which are mainly controlled by the transporter Ank and ectonucleotide pyrophosphatase/phosphodiesterase 1 (ENPP1). We demonstrated previously that transforming growth factor (TGF)-β1 increased ePPi production by inducing Ank and Enpp1 expression in chondrocytes. As the TGF-β1 level raises in synovial fluid under hypoxic conditions, we investigated whether hypoxia may transform SM as a major source of ePPi production. Synovial fibroblasts and SM explants were exposed to 10 ng/mL of TGF-β1 in normoxic or hypoxic (5% O_2_) culture conditions. Ank and Enpp1 expression were assessed by quantitative PCR, Western blot and immunohistochemistry. ePPi was quantified in culture supernatants. RNA silencing was used to define the respective roles of *Ank* and *Enpp1* in TGF-β1-induced ePPi generation. The molecular mechanisms involved in hypoxia were investigated using an *Ank* promoter reporter plasmid for transactivation studies, as well as gene overexpression and RNA silencing, the respective role of hypoxia-induced factor (HIF)-1 and HIF-2. Our results showed that TGF-β1 increased Ank, Enpp1, and therefore ePPi production in synovial fibroblasts and SM explants. Ank was the major contributor in ePPi production compared to ENPP1. Hypoxia increased ePPi levels on its own and enhanced the stimulating effect of TGF-β1. Hypoxic conditions enhanced *Ank* promoter transactivation in an HIF-1-dependent/HIF-2-independent fashion. We demonstrated that under hypoxia, SM is an important contributor to ePPi production in the joint through the induction of *Enpp1* and *Ank*. These findings are of interest as a rationale for the beneficial effect of anti-inflammatory drugs on SM in crystal depositions.

## 1. Introduction

Chondrocalcinosis (CCA) is a common joint disease described by the deposition of calcium-containing crystals, mostly calcium pyrophosphate dihydrate (CPPD) inside articular cartilage and synovial membrane [1]. Its frequency increases with age, and its prevalence can reach 20–30% over the age of 80 and exceed 30% after the age of 90 [2]. Sporadic or idiopathic forms represent the vast majority of CCA [3]. CPPD deposition is also found, though at a lower frequency, in patients with osteoarthritis (OA) [4] or rheumatoid arthritis (RA) [5]. A variable level of hypoxia occurs in the synovial membrane (SM) of OA [6] and RA patients [6,7], depending on inflammation and joint pressure increase during joint mobilization. Thus, higher lactate levels (a biological signature of hypoxia) were found in the synovial fluid from acute CCA patients compared to patients with chronic CCA or OA [8]. Moreover, in an experimental model of knee joint mechanical instability in rabbits, synovitis was responsible for hypoxia and hypercapnia in the synovial fluid of the injured animals [9]. These observations underline the occurrence of hypoxia in the SM and its possible contribution to progression of rheumatic diseases with or without microcrystals.

A regional production of extracellular inorganic pyrophosphate (ePPi), the anionic component of CPPD crystals, confirms CPPD formation [10]. Interestingly, increased ePPi levels were found in the synovial fluid [11] and plasma [12] of patients with OA or CCA. Two proteins are mainly responsible for the production of ePPi: the ectonucleotide pyrophosphatase/phosphodiesterase 1 (ENPP1), whose activity was detected in synovial fluid [13], and the transmembrane transporter Ank, which is the main contributor of ePPi production by articular chondrocytes [14]. However, the accumulation of ePPi could also result from its reduced degradation. Indeed, tissue non-specific alkaline phosphatase (TNAP) activity, which was also detected in synovial fluid [15], induces the cleavage of ePPi into two molecules of extracellular inorganic phosphate (ePi). Interestingly, this activity mainly originates from blood vessels in the synovial subliming layer [16], as no alkaline phosphatase activity was described to date in the intimal lining layer (containing the synovial fibroblast cell population). Therefore, the balance between ePPi production and degradation in the synovial fluid strongly depends on these three proteins ENPP1, Ank, and TNAP. However, their respective contribution in the SM remains unclear, especially under hypoxic conditions.

Transforming growth factor-β1 (TGF-β1) has been shown to be the key inducer of ePPi production by chondrocytes [10]. Former studies have proven that *Ank* mRNA level was greater in human chondrocytes displayed to TGF-β1 compared to controls [17], just like murine cartilage [18]. Moreover, TGF-β1 is able to increase the Ank protein in a rat sarcoma virus/rapidly accelerated fibrosarcoma-1/extracellular-signal-regulated kinase (RAS/RAF-1/ERK)-dependent pathway in chondrocytes [14]. Ank was also shown to be regulated by hypoxia in growth plate chondrocytes [19], whereas TGF-β1 levels were found to be increased in synovial fluid under hypoxic conditions with subsequent enhanced vascular endothelial growth factor (VEGF) secretion by synovial fibroblasts [20]. Nevertheless, although TGF-β1 levels were demonstrated to be greatly elevated in synovial fluids of OA patients with CPPD [21], the effect of TGF-β1 on ePPi generation by SM under hypoxic conditions is not well documented.

Zaka et al. showed that Ank and ePPi production is modulated by growth plate chondrocytes [19]. Our team has previously shown the important part of TGF-β1 in ePPi production by chondrocytes. ePPi has a protective role for cartilage by maintaining the differentiated phenotype of articular chondrocytes [14,22]. As acute or chronic pro-inflammatory episodes lower the O_2_ level in the synovial fluid and TGF-β1 influences the cartilage (which is in contact with this fluid), we were interested in both the impact of hypoxia and TGF-β1 on the SM which produces this fluid. If ePPi production increases, it can lead to the formation of CPPD crystals by binding calcium; the latter being themselves pro-inflammatory. It is therefore possible that an inflammatory crisis can generate the formation of CPPD crystals in the SM and then the cartilage. We hypothesized that *Ank* expression and ePPi production were upregulated by hypoxia and TGF-β1 in the SM and synovial fibroblasts. To verify our hypothesis, we first studied in the SM and synovial fibroblasts the influence of hypoxia on *Ank* expression and ePPi production, then the effect of TGF-β1, and finally, the combined effect of hypoxia and TGF-β1.

The purpose of this study was to investigate how hypoxia and TGF-β1 influence ePPi production by the SM and/or synovial fibroblasts.

## 2. Results

### 2.1. TGF-β1 Stimulates the Expression of Ank and Enpp1 to Increase the Production of ePPi and ENPP1 Activity by Synovial Fibroblasts

Our preliminary experiments confirmed that in our experimental conditions, synovial fibroblasts keep their B-type mature phenotype by strongly expressing the specific markers synoviolin and hyaluronic acid synthase 1 and 2 [23]. This also confirmed the lack of contamination by macrophage-like cells. We examined the time course of Ank and Enpp1 mRNA expression in TGF-β1-stimulated synovial fibroblasts (Figure 1A). Ank and Enpp1 were up regulated from 3 h. They reached a peak value at 24 h after TGF-β1 exposure to increase respectively by around 3-fold and 6.5-fold. Western blotting confirmed that Ank was induced from 12 h after TGF-β1 challenge, whereas Enpp1 was upregulated after 6 h (Figure 1B). Moreover, as shown in Figure 1C, ePPi level increased continuously, from 2-fold after 6 h of stimulation with TGF-β1 until 6-fold at 48 h. In these experimental conditions, TGF-β1 stimulated Enpp1 activity by 3-fold at 24 h (Figure 1D), whereas we failed to detect any alkaline phosphatase (APase) activity (tissue specific and tissue non-specific). Taken together, these data demonstrated that Ank and Enpp1 were induced by TGF-β1, thus resulting in an increased ePPi production and Enpp1 activity by synovial fibroblasts.

### 2.2. Ank Contributes More Than Enpp1 to the TGF-β1-Induced Production of ePPi by Synovial Fibroblasts

The siRNA technology was used to investigate the respective contributions of Ank and Enpp1 to TGF-β1-induced production of ePPi by synovial fibroblasts cultured in monolayer. Control experiments showed that siRNAs were efficient, as they reduced targeted mRNA level by more than 80% compared to non-silencing scramble siRNA in basal and reduced the contribution of the target gene to the stimulating effect of TGF-β1 by about 50% (Appendix A) at the time of maximal gene expression. No effect was observed with scramble siRNA on ePPi production (Appendix A). When ePPi levels were measured in culture supernatant of monolayer transfected with siRNA, inhibition of Ank and Enpp1 reduced the basal level by 50% and 40% respectively and accounted for a 60% and 50% decrease in TGF-β1-stimulated cells respectively (Appendix A). These data demonstrated that Ank contributed slightly more than Enpp1 to the regulation of ePPi level in both resting and TGF-β1-stimulated synovial fibroblasts.

### 2.3. Hypoxia Enhances the Production of ePPi by Synovial Explants

Ex vivo explants cultures of SM were submitted to RNA silencing, as presented in the previous section, to investigate the contributions of Ank and Enpp1 to TGF-β1-induced production of ePPi when exposed to normoxia (21% O_2_) and relative hypoxia (5% O_2_). Exposure to low oxygen concentration led to an increase in Ank and Enpp1 mRNA expression, as shown in Figure 2A,B. This was accompanied by an important increase in ePPi level (Figure 2C). These effects were similar as those obtained in synovial fibroblasts cultured in monolayer under hypoxia.

As observed for monolayers cultured in normoxia (Appendix A), siRNAs were also efficient ex vivo, as they reduced Ank and Enpp1 mRNA levels by more than 80% compared to the scramble siRNA in basal conditions. In addition, they efficiently reduced the stimulating effect of TGF-β1 by about 50% in normoxia and 80% in hypoxia (Figure 2A,B). No effect was observed with scramble siRNA on TGFβ1-induced ePPi production (Figure 2C). When ePPi levels were measured in supernatant of transfected SM explants, the inhibition of Ank and Enpp1 reduced basal ePPi level in both normoxia and hypoxia (50% and 40% respectively). Moreover, inhibition of Ank and Enpp1 reduced the TGF-β1-induced ePPi production in normoxia (70% and 60% respectively) and accounted for an 85% and 75% respective decrease in ePPi level in hypoxia (Figure 2C).

Immunohistochemistry (IHC) analyses of SM explants demonstrated for Ank (Figure 3A), and, to a lesser extent, for Enpp1 (Figure 3B), an increased expression of these proteins in tissue under hypoxia for 48 h compared to normoxia (Figure 3C). In addition, ePPi level increased by 3-fold under hypoxia (Figure 3D), while ENPP1 activity decreased by around 30% (Figure 3E).

### 2.4. Induction of Ank by Hypoxia Is Mainly a HIF-1 Dependent Event in Synovial Fibroblasts

Control experiment with pGL3-hypoxia-induced factor (HIF)-response elements (HRE)-Luc confirmed the effectiveness of hypoxia in our experimental system as it induced an 8-fold increase in luciferase activity from 5% O_2_ (Figure 4B). Using pGL3-Ank-Luc construction, we demonstrated the increase in Ank reporter activity by low O_2_ tension with the following rank order: 1% > 3% > 5% (Figure 4A). The overexpression of HIF-1 or HIF-2 increased by 28- to 36-fold the transactivation of pGL3-HRE-Luc in hypoxia experiments (Figure 4D). Using these constructs, we demonstrated the main role of HIF-1 in Ank regulation by hypoxia (Figure 4C) as a 25-fold increase was observed in transfected cells overexpressing HIF-1, while only an 8-fold increase was seen for cells overexpressing HIF-2.

When synovial fibroblasts were transfected with scramble siRNA, no change in Ank promoter and HRE transactivations could be detected (Figure 5A,B). Hif-1 and Hif-2 siRNAs yielded a comparable inhibitory effect (Figure 5B). Control experiments demonstrated the efficacy of these siRNA sequences when assessing HIF-1 or HIF-2 mRNA expression (Appendix A). Using Hif-1 siRNA, the induction of Ank promoter by hypoxia was almost suppressed (Figure 5A), whereas Hif-2 siRNA provided only a discrete inhibition. Interestingly, we observed an increase in Hif-1 mRNA level in the presence of Hif-2 siRNA, and vice-versa (Appendix A). This compensatory mechanism between HIF-1 and HIF-2 may explain the minor effect of Hif-2 invalidation. The observations made with Ank promoter were confirmed at the mRNA level, as the hypoxia-induced increase in Ank expression was nearly completely cancelled by Hif-1 siRNA, while Hif-2 siRNA remained ineffective (Figure 5C). The results were similar for ePPi release (Figure 5D).

## 3. Discussion

We demonstrate here the important role of TGF-β1 and hypoxia in ePPi generation by synovial tissue. We show that the ePPi production depended more on ANK than on ENPP1, and that it was increased in the context of hypoxia. The *Ank* promoter was mainly transactivated by HIF-1 but not by HIF-2 in this context.

We previously demonstrated that TGF-β1 induced ePPi production by articular chondrocytes, and that this was mainly due to Ank contribution over Enpp1 (around 70% over 30%) [14]. In the present study, we show a more balanced system in synovial fibroblasts and SM explants with contributions around 55% over 45%, respectively. Enpp1 was described to contribute between 35% and 50% to ePPi production by osteoblasts [24]. Altogether, these observations suggest that ePPi production in the whole joint is mainly dependent on Ank expression over Enpp1.

A striking observation was the major influence of hypoxia on ePPi production by SM explants. Indeed, the hypoxic condition strongly increased *Ank* and *Enpp1* mRNA expression, thus resulting in an increase in ePPi production. The influence of hypoxia on Ank and ePPi production was described previously in chondrogenic cells though with growth plate phenotype [19]. In this study, Zaka et al. observed an opposite phenomenon, as hypoxia decreased Ank expression as well as ePPi levels. Enpp1 regulation by hypoxia has not yet been documented to our knowledge. However, our experiments of *Enpp1* invalidation are concordant with our demonstration of the inducing effect of hypoxia on ePPi production and on Ank expression.

There are two HREs sensitive to hypoxia on the *Ank* gene promoter. This explains the transregulation of the *Ank* promoter elements by HIF proteins as shown by Zaka et al. [19]. The authors described that HIF-1 efficiently reduced the transactivation of *Ank* promoter during a hypoxic response [19], whereas we show here that HIF-1 was involved in the activation of *Ank* promoter during the hypoxic response. Our team has previously demonstrated the mechanisms of *Ank* regulation (mainly through ERK pathway) by TGF-β1 in ePPi production by chondrocytes [14]. Hypoxia has also been shown to induce ERK pathway as well as other MAPKs in cartilage [25]. Other pathways such as phosphoinositide 3-kinase (PI3K)/Akt/forkhead box O (FOXO) are also are also activated by hypoxia in cartilage and could possibly explain the modulation of the *Ank* gene [26].

A technical difference may explain this discrepancy, as Zaka et al. used the *Ank* proximal promoter (around 1 kb) [19]. However, in our work, we use a longer 2.7 kb promoter, which includes an E26 transformation-specific Like-1 (*Elk-1*) binding site, which is a strong inducer of *Ank* promoter in TGF- β1-stimulated condition. Moreover, Zaka et al. demonstrated that differentiated embryo-chondrocyte expressed gene 1 (*Dec1*) acted as a transcriptional co-repressor of the *Ank* promoter [19]. In our cell system, *Dec1* expression was barely detectable, in both normoxic and hypoxic conditions. These observations may explain the inducing effect of HIF-1 in cells derived from the SM.

Zaka et al. also demonstrated by IHC that Ank expression was higher in tissues where oxygen tension is higher, i.e., hypertrophic cartilage and SM [19]. However, the influence of hypoxia on the SM is missing in Zaka’s work. It is also possible that hypoxia may exert a different influence depending on the tissues (or the chondrocyte phenotypes), i.e., on SM and articular cartilage. Indeed, in its physiological state, articular cartilage is a hypoxic tissue, while SM or growth plate cartilage are normoxic tissues. Therefore, hypoxia represents a stressor for the SM, which may explain the differential regulation of Ank expression.

The differential influence of hypoxia depending on the tissue was already reported at the molecular level, as HIF-1 is often described as a VEGF inducer in endothelial cells [27,28,29], and also in chondrogenic cells [19]. However, another work on the brain microvasculature supported the contrary, since a decreased VEGF expression was reported, while Hif-1 mRNA accumulated [30].

The absence of any clear influence of HIF-2 on *Ank* promoter compared to HIF-1 may be due to the expression pattern of these transcription factors. Indeed, it was shown that expression of HIF-1 was correlated with a lower HIF-2 level and vice-versa [31]. However, we show in this present work that the mRNA level of both Hif-1 and Hif-2 was high in hypoxia.

Thus, the difference between HIF-1 and HIF-2 may originate from the molecular level. Even though these transcription factors share the same HRE binding sites [32], they can interact with different proteins to form, or not, a translational complex. Indeed, HIF-1 was shown to bind pyruvate kinase M2 (PKM2) in a translational complex promoting the hypoxic response, while HIF-2 was unable to bind PKM2 [33]. This may be the case in the synovial tissue, where possibly only HIF-1 could be active on *Ank* promoter. As our results are obtained with the culture of rat primary synoviocytes, they do not necessarily represent what happens in the SM in vivo and even less in human tissues. In vivo studies in rats and in vitro studies in humans are required to confirm the observed effects.

The drop in joint oxygen levels results in a very high Ank expression and ePPi production, which can induce the formation of CPPD crystals and cause episodes of secondary chondrocalcinosis [34]. The latter being also found in OA, they can also be responsible or trigger its more rapid progression [35]. Interestingly, such a link between hypoxia and a higher Ank expression is in line with clinical findings, where high lactate levels were found in synovial fluid from patients with acute CPPD compared to patients with OA [8]. Indeed, lactates reflect, at least in part, a hypoxic environment, while ANK seems to be the main contributor to ePPi generation. Moreover, it was shown that hypoxia blocks the spontaneous calcifications in bone marrow-derived mesenchymal stem cells cultures [36], as well as in pre-osteoblasts cultures [37]. Such calcifications mainly consist of hydroxyapatite depositions, which are antagonized by elevated ePPi levels. Thus, our data showing that hypoxia induces Ank and Enpp1 expression are also in line with these findings, as they support the hypothesis of ePPi antagonizing extracellular inorganic phosphate depositions. This is what happens in *ank*/*ank* deficient mice, which have severe ubiquitous hydroxyapatite depositions [29].

## 4. Materials and Methods

Reagents were obtained from Merck (Saint-Quentin-Fallavier, France) and media components by Lonza (Colmar, France), unless specified otherwise.

### 4.1. Study Design

First, we studied the kinetics of Ank and Enpp1 expression in response to a TGF-β1 (10 ng/mL) challenge in rat synovial fibroblast, at the mRNA and protein level. Enzymatic activities (Enpp1 and APase) were also assessed on total cell protein extracts 24 h after TGF-β1 stimulation. ePPi levels were determined in culture supernatants until 48 h after TGF-β1 challenge.

Second, we investigated the contribution of Ank and Enpp1 to ePPi production and Enpp1 activity, in both rat synovial fibroblasts and SM explants in the presence or absence of TGF-β1, and under normoxia (21% O_2_) or hypoxia (5% O_2_). For cells, we also used silencing double stranded RNAs, known as small interfering RNAs (siRNAs), against *Ank* or *Enpp1*. For explants, Ank and Enpp1 protein expression was assessed by IHC.

Thirdly, to decipher the influence of hypoxia on *Ank* expression and ePPi production, we used a 2.7 kb construct of *Ank* promoter in a Firefly luciferase reporter assay as previously described [22]. We tested the influence of hypoxia on *Ank* promoter transactivation and assessed hypoxia efficacy with an HRE-Luc (luciferase reporter of HIF activity) plasmid. We tested three different hypoxia levels, by setting O_2_ at 1, 3, or 5%.

Finally, we checked for the contribution of the transcription factors HIF-1 and HIF-2, which are the most characterized isoforms of the hypoxic response, on the transactivation of *Ank* promoter. For that purpose, we used either overexpression plasmids or silencing siRNAs directed against these factors.

### 4.2. Culture of SM Explants and Synovial Fibroblasts

Synovial tissues were obtained from Wistar male rats (130–150 g; Charles River, Saint-Aubin-les-Elbeuf, France), euthanized under universal anesthesia (AErrane™, Baxter SA, Maurepas, France) according to European animal care guidelines, after approval by our internal ethics committee. Synovial fibroblasts were obtained by chronological digestion with pronase and collagenase as described previously [38]. The cells were washed twice in phosphate buffered saline (PBS) and cultured until 70% confluence in 25 cm^2^ flasks at 37 °C in a humidified atmosphere containing 5% CO_2_. The synovial fibroblasts were exploited between passages 3 and 6, to avoid contamination by synovial macrophages and to ensure a fibroblastic morphology [38]. SM explants were cultured in the same experimental conditions. For hypoxic cultures conditions, synovial fibroblasts or SM explants were incubated in 1, 3, or 5% of oxygen in an incubator.

### 4.3. RNA Isolation and Reverse Transcription–Quantitative PCR (RT–qPCR)

SM explant or synovial fibroblast mRNA levels were determined using SYBR Green-based qPCR. Total RNAs were extracted using the Nucleospin RNA kit^®^ (Macherey Nagel, Hoerdt, Germany). A total of 100 nanograms of total RNAs were reverse transcribed at 37 °C for 90 min in a 20 μL reaction mixture with 200 U Moloney Murine Leukemia Virus reverse transcriptase (Invitrogen, Fisher Scientific, Illkirch, France), 1.5 mM MgCl_2_, 5 μM random hexamer primers, and 10 mM dNTP. RT-PCR was accomplished using Step One Plus^TM^ (Applied Biosystems, Fisher Scientific, Illkirch, France) technology with specific primers (Table 1) and iTAQ SYBRgreen^TM^ master mix system (Bio-Rad, Steenvoorde, The Netherlands). The specific PCR products melting temperature was determined using a melting curve. The mRNA levels of the gene of interest and of the ribosomal protein 29 (*RP29*) were quantified in parallel for each sample using the ΔΔCt method. Finally, the results were displayed as fold expression over the control.

### 4.4. Western Blot Analysis

Rat synovial fibroblasts were harvested and lysed in 1X Laemmli buffer. Protein samples were run on an SDS-polyacrylamide gel (10%) and transferred onto a polyvinylidene fluoride membrane [14]. After 2 h in blocking buffer (tris-buffered saline-Tween^®^ 20 [TBST], 5% non-fat dry milk), membranes were cleaned three times with TBST and incubated overnight at 4 °C with primary antibodies. The rabbit polyclonal antibodies against Ank and Enpp1 (Eurogentec, Angers, France) were 1/500 dilution [14]. Antibody against β-actin was 1/8000. After three washings with TBST, each blot was incubated for 1 h at room temperature with anti-rabbit IgG conjugated with horseradish peroxidase (HRP) (Cell Signaling Technology, Ozyme, Saint-Cyr-L’École, France) at 1/2000 dilution. Protein bands were detected by chemiluminescence.

### 4.5. Enpp1 and APase Activities

Rat synovial fibroblasts or SM were collected and lysed in a buffer containing 1% Triton X-100, 1.6 mM MgCl_2_, and 0.2 M Tris base (pH 8.1). Total protein extracts (quantified by bicinchoninic acid assay) were incubated for 15 min with 1 μmol of p-nitrophenylthymidine 5′-monophosphate for Enpp1 activity or for 2.5 h with 5 μmol of p-nitrophenyl phosphate for APase activity; these enzymatic activities both generate p-nitrophenol. At the end of incubation, the reaction was stopped up by adding 10 μmol of ethylenediaminetetraacetic acid (EDTA) and 200 μmol of NaOH. Absorbance was read at 410 nm. The calibration curves, ranging from 0 to 0.2 mM p-nitrophenol, were contained in each assay. Results were expressed as units per milligram of total cell proteins.

### 4.6. Radiometric Assay for ePPi

ePPi levels were measured in culture supernatants of synovial cell monolayers and SM explants, using the differential adsorption between uridine diphospho-(6-^3^H) glucose (PerkinElmer, Villebon-sur-Yvette, France), and its reaction product 6-phospho-(6-^3^H) gluconate on activated charcoal [39]. The standards, ranging from 10 to 400 pmol of ePPi, were included in each assay. After adsorption of the reaction mixture on charcoal and centrifugation at 16,000× *g* for 10 min, 100 μL of the supernatant was counted for radioactivity in 5 mL of Bio-Safe II (Research Products International Corp., Mt. Prospect, IL, USA). Results were expressed as pmol of ePPi per microgram of total cell proteins.

### 4.7. Immunohistochemistry

SM were fixed for 24 h in 4% paraformaldehyde then embedded in paraffin. Paraffin sections (5 μm thick) were deparaffinized in Tissue Clear (Bayer Diagnostics, Puteaux, France) and rehydrated in ethanol. Hydrogen peroxide (3%) was then added to quench endogenous peroxidases, and specimens were incubated overnight in a pH 6.0 citrate buffer at 70 °C. Nonspecific sites were blocked for 1 h with ImmPRESS™ Kit (MP-7500; Vector Laboratories, Eurobio Scientific, Les Ulis, France). Then sections were incubated overnight in a humidified atmosphere at 4 °C with the different primary polyclonal antibodies. Antibodies were used at 1/200 and 1/500 for Ank and Enpp1. HRP-labelled secondary antibodies (goat anti-rabbit IgG [Vectastain ABC kit; Novocastra, Le Perray en Yvelines, France] were applied for 30 min. The signal was obtained with permanent AEC Chromogen^®^ (Diagomics, Blagnac, France). Counterstaining of nuclei was performed with Harris hematoxylin, and slices were mounted in Eukitt^®^ (CML, Nemours, France). Images were acquired using DMD108 micro-imaging device, connected to a 3-million-pixel charge-coupled device camera (Leica Microsystems, Nanterre, France) and software Leica Acquisition Suite (LAS V3.3). Magnification was 10X and numerical aperture was 0.25.

### 4.8. Silencing Experiments Using siRNA

The double-stranded sequences for the non-silencing scramble siRNA (Scr; medium GC content) and the siRNAs targeting *Ank*, *Enpp1*, *Hif-1,* and *Hif-2* (Table 2) were designed by and acquired from Eurogentec (Angers, France). Transfections were conducted with each siRNA at 10 nM, using INTERFERin^TM^ (Polyplus-transfection^®^ SA, Illkirch-Graffenstaden, France). siRNAs were diluted in serum-free medium, INTERFERin^TM^ was then added for a short incubation at room temperature. Cells were washed with PBS and placed in serum-free medium. The siRNA-INTERFERin^TM^ mix was then added to the cell culture for 12 h. Then, cells were stimulated or not with 10 ng/mL TGF-β1 under normoxic (21% O_2_) or hypoxic (5% O_2_) conditions for 48 h. Knockdown efficiencies were quantified at the mRNA level for each targeted gene and using reporter assay in synovial fibroblasts transfected with HRE-Luc construct.

### 4.9. Transient Transfection

Plasmid pmaxGFP™ (Amaxa^®^, Lonza Cologne AG, Köln, Germany), encoding a green fluorescent protein, was used to determine the transfection efficiency. To assess their activity, Ank-Luc (Firefly luciferase reporter of Ank promoter activity) and HRE-Luc were cloned in pGL3 plasmid vector (Promega, Charbonnières-les-Bains, France) to produce pGL3-Ank-Luc and pGL3-HRE-Luc, respectively. To be overexpressed, Hif-1 and Hif-2 were cloned in pcDNA3.1 plasmid vector (Invitrogen^TM^, Thermo Fisher Scientific, lllkirch-Graffenstaden, France) to produce pcDNA3.1-HIF-1 and pcDNA3.1-HIF-2, respectively. Twenty-four hours before transfection, rat synovial fibroblasts were seeded in 6-well plates at 5 × 10^5^ cells/well and grown to 60–80% of confluence. Cells were transfected with 500 ng of pGL3-Ank-Luc or pGL3-HRE-Luc and 10 ng of pCMV-Renilla (Promega, Charbonnières-les-Bains, France) as internal control. The transfection was performed using 10 µL of polyethylenimine reagent (Euromedex, Souffelweyersheim, France) in 1 mL of complete medium. Twenty-four hours later, the reagent-containing medium was replaced by complete growth medium. Then, cells were stimulated with 10 ng/mL TGF-β1 in normoxia (21% O_2_, used as oxygenation standard control) or hypoxia (1, 3 and 5% O_2_) for 48 h. Cells were transfected as described above an used as experimental control and in addition, they were also transfected or not with 500 ng of empty pcDNA3.1 (used as expression control), or pcDNA3.1-HIF-1, or pcDNA3.1-HIF-2. Then, cells were handled as mentioned above. After 12 h of incubation, cells were stimulated with 10 ng/mL TGF-β1 in hypoxia (5% O_2_) for 48 h.

### 4.10. Reporter Gene Assay

Luciferase was assessed using Dual-Luciferase Reporter Assay^TM^ kit (Promega, Charbonnières-les-Bains, France) according to the manufacturer’s instruction. Briefly, rat synovial fibroblasts transfected with reporter gene constructs were stimulated or not for 48 h with TGF-β1. Cells were harvested in lysis buffer for 15 min at room temperature. Cell lysates were put in Firefly luciferase substrate and luminescence was glance at with GloMax^®^ 96 Microplate Luminometer (Promega, Charbonnières-les-Bains, France). Then, Renilla luciferase substrate was added to the samples and luminescence was acquired by the same method. Results were expressed as the mean ratio of Firefly/Renilla luciferase activity, in fold induction over control.

### 4.11. Statistical Analysis

Results are expressed as the mean ± standard deviation (SD) with a minimum of 3 independent experiments (*n* = 3). Statistical analyses were performed using non-parametric. Comparisons were made by analysis of variance, followed by Fisher’s t post hoc test, using the GraphPad PrismTM 6.0 software (GraphPad Software, San Diego, CA, USA). A value of *p* < 0.05 was considered significant.

## 5. Conclusions

In conclusion, we demonstrate that in the presence of TGFβ-1, the synovial membrane, is an important contributor to ePPi production in the joint though the induction of the *Ank* gene. Moreover, this is also the case in another pathophysiological condition such as hypoxia, where *Ank* expression is mainly induced by HIF-1 but not HIF-2. This mechanistic insight into ePPi production under hypoxic conditions provides a foundation for potential therapeutic targets in conditions related to CPPD crystal formation. This work adds to the realms of rheumatology and pharmaceutical development, shedding light on the pathophysiological mechanisms underlying common yet debilitating conditions like OA and CPPD crystal deposition disease. Our findings are of interest as a rationale for beneficial effect of anti-inflammatory drugs on SM in crystal depositions.

## Figures and Tables

**Figure 1 pharmaceuticals-17-00738-f001:**
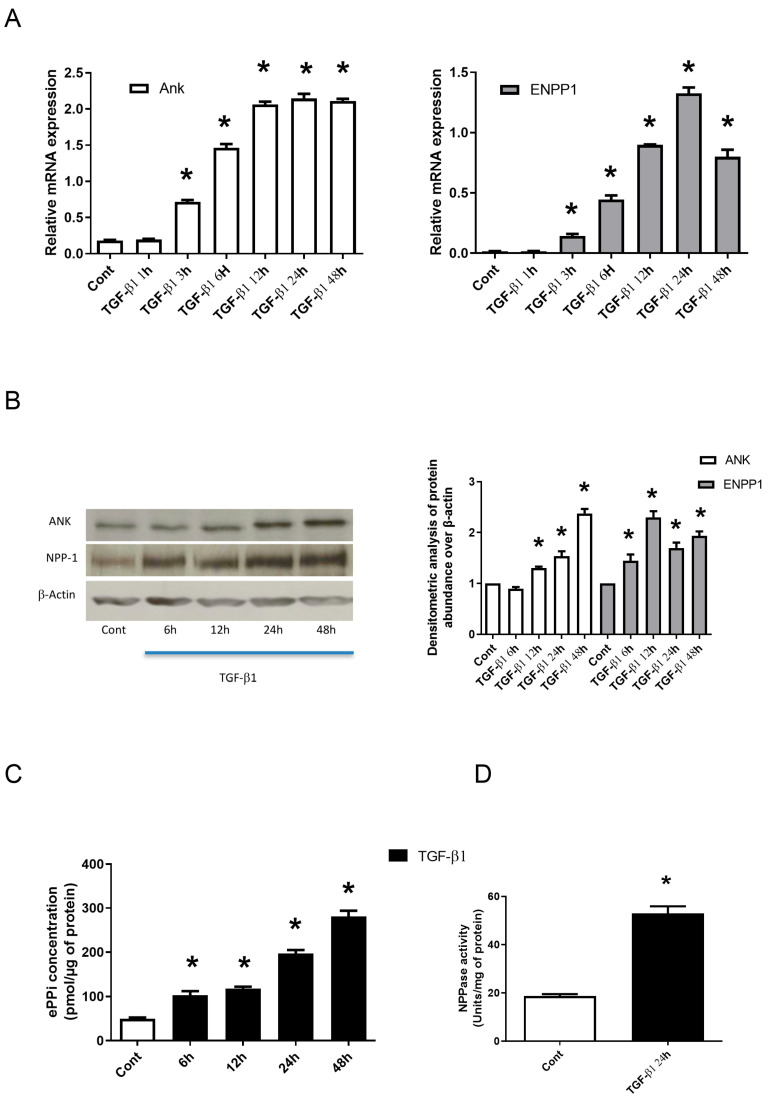
TGF-β1 stimulates the expression of *Ank* and *Enpp1* to increase the production of ePPi and Enpp1 activity by synovial fibroblasts in vitro. (**A**–**D**) Synovial fibroblasts cultured in monolayer were exposed or not to 10 ng/mL of TGF-β1. (**A**) Kinetics from 1 to 48 h of *Ank* and *Enpp1* expression (mRNA). Total RNA was extracted from synovial fibroblasts and subjected to RT-qPCR analysis. The abundance of *Ank* (left panel) and *Enpp1* (right panel) mRNA was normalized to that of *Rp29* mRNA. Results are expressed as means (±SD) over *Rp29* values. (**B**) Kinetics from 6 to 48 h of ANK and ENPP1 expression (protein). Total proteins were extracted from synovial fibroblasts and subjected to Western blotting using polyclonal anti-Ank and anti-Enpp1 antibodies. Band intensities (left panel) were quantified by densitometry (right panel). The abundance of these proteins was normalized to that of β-actin protein and expressed as induction folds over control value. (**C**) ePPi level was assayed radiometrically using supernatant of synovial fibroblasts cultured from 6 to 48 h and normalized to the amount of total cell proteins. Data are expressed as mean (±SD) in picomoles per microgram of protein. (**D**) Enpp1 activity. Proteins were extracted from synovial fibroblasts cultured for 24 h. Enzyme activity was standardized to the total cell proteins. Results are stated as mean (±SD) in micromoles of paranitrophenol per minute per milligram of protein. Statistically significant differences from the control are indicated as * for *p* < 0.05. (*Ank*: inorganic pyrophosphate transport regulator; Cont: control condition meaning no TGF-β1 treatment; *Enpp1*: ectonucleotide pyrophosphatase/phosphodiesterase 1; ePPi: extracellular inorganic pyrophosphate; *Rp29*: ribosomal protein 29; RT-qPCR: reverse transcription–quantitative polymerase chain reaction; SD: standard deviation; TGF: transforming growth factor).

**Figure 2 pharmaceuticals-17-00738-f002:**
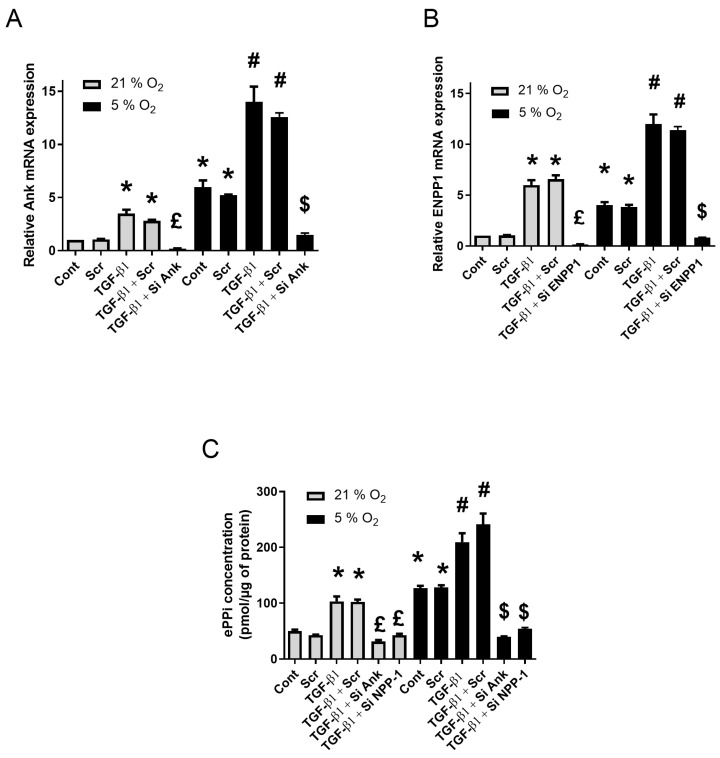
Hypoxia increases *Ank* and *Enpp1* mRNA levels and the production of ePPi ex vivo. (**A**–**C**) SM explants were transfected or not with scramble siRNA or siRNAs directed against *Ank* or *Enpp1* for 12 h, before being challenged with 10 ng/mL of TGF-β1 for 48 h either in normoxia (21% O_2_) or hypoxia (5% O_2_). Total RNA was extracted from explants and then subjected to RT-qPCR. The abundance of *Ank* (**A**) and *Enpp1* (**B**) mRNA was normalized to that of *Rp29* mRNA. Results are expressed as means (±SD) over *Rp29* values. (**C**) ePPi levels were assessed in culture supernatants of SM explants. They were normalized to the amount of total cell proteins and are expressed as means (±SD) in picomoles per microgram of protein. Statistically significant differences are indicated as: * for *p* < 0.05 from the control in normoxia, # for *p* < 0.05 from control in hypoxia, £ for *p* < 0.05 from TGF-β1-stimulated cells in normoxia, and $ for *p* < 0.05 from TGF-β1-stimulated cells in hypoxia. (*Ank*: inorganic pyrophosphate transport regulator; Cont: control condition meaning no TGF-β1 treatment and no transfection; *Enpp1*: ectonucleotide pyrophosphatase/phosphodiesterase 1; ePPi: extracellular inorganic pyrophosphate; *Rp29*: ribosomal protein 29; RT-qPCR: reverse transcription–quantitative polymerase chain reaction; Scr: scramble RNA meaning non-silencing RNA; SD: standard deviation; Si: means siRNA or small interfering RNA; SM: synovial membrane; TGF: transforming growth factor).

**Figure 3 pharmaceuticals-17-00738-f003:**
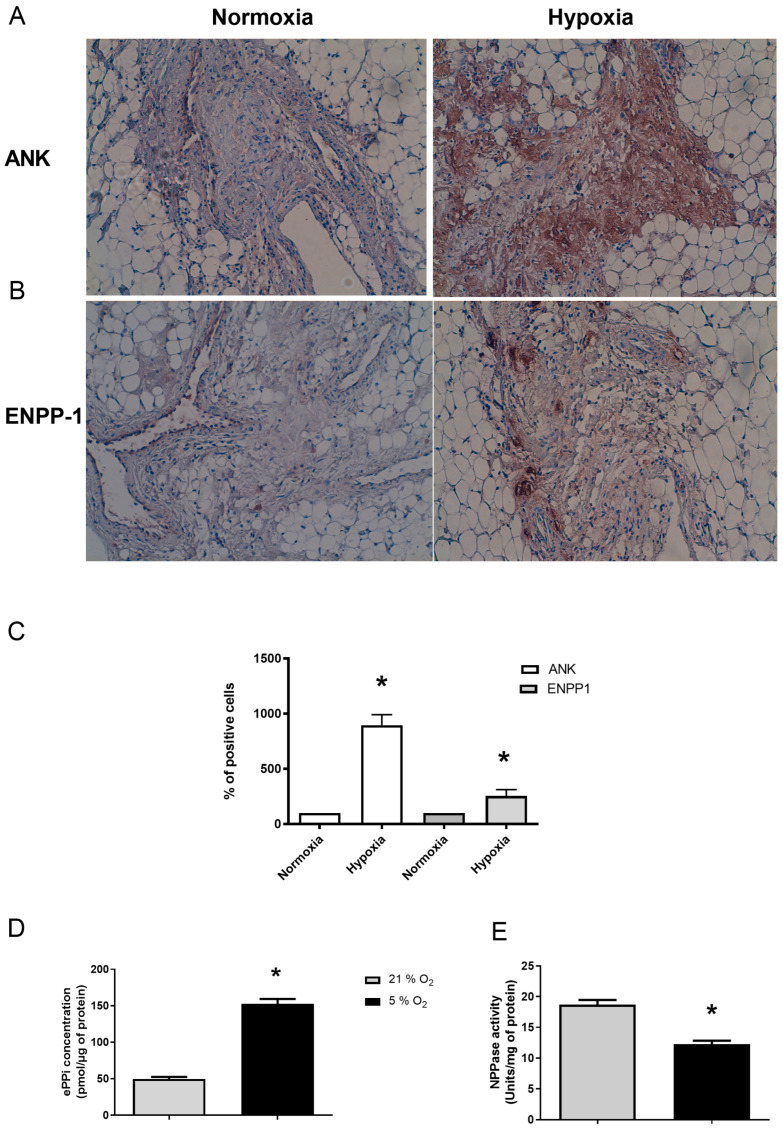
Hypoxia enhances Ank and Enpp1 protein expression to increase ePPi production and Enpp1 activity ex vivo. (**A**–**E**) SM explants were cultured either in normoxia (21% O_2_) or hypoxia (5% O_2_) for 48 h. (**A**,**B**) Samples were subjected to immunohistochemistry using polyclonal anti-Ank (**A**) and anti-Enpp1 (**B**) antibodies. The exhibited pictures are representative of at least three independent experiments. The tissue presence of proteins is characterized by a brown staining and the counterstaining of cell nuclei appear as blue. Magnification is ×10. (**C**) The percentage of positively stained cells in SM was quantified by two independent observers and results are expressed as mean (±SD). (**D**) ePPi levels were assessed in culture supernatants of SM explants and normalized to the amount of total cell proteins. Results are expressed as means (±SD) in picomoles per microgram of protein. (**E**) Enpp1 activity. Proteins were extracted from SM. Enzyme activity was normalized to the amount of total cell proteins. Results are expressed as mean (±SD) in micromoles of paranitrophenol per minute per milligram of protein. Statistically significant differences are indicated as * for *p* < 0.05 from the control in normoxia. (Ank: inorganic pyrophosphate transport regulator; Enpp1: ectonucleotide pyrophosphatase/phosphodiesterase 1; ePPi: extracellular inorganic pyrophosphate; SD: standard deviation; SM: synovial membrane).

**Figure 4 pharmaceuticals-17-00738-f004:**
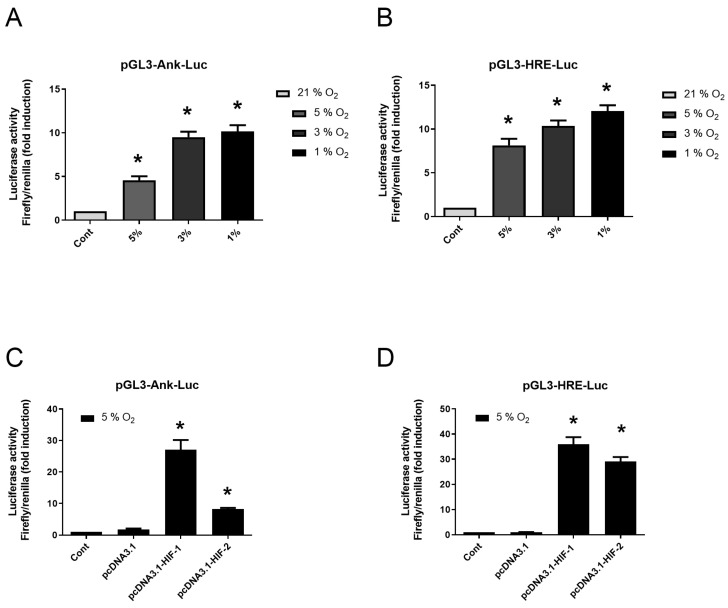
Hypoxia increases the transactivation of *Ank* promoter in vitro. (**A**,**B**) Synovial fibroblasts were transfected or not for 24 h then cultured either in normoxia (21% O_2_) or hypoxia (various oxygen percentage). (**A**) Synovial fibroblasts were transfected or not with pGL3-Ank-Luc and pCMV-Renilla reporter. (**B**) Control of hypoxia efficiency using synovial fibroblasts transfected or not with pGL3-HRE-Luc and pCMV-Renilla reporter. (**C**,**D**) Synovial fibroblasts were transfected or not then cultured in hypoxia (5% O_2_). (**C**) Synovial fibroblasts were transfected or not with pcDNA3.1, or pcDNA3.1 construct to overexpress HIF-1 or HIF-2, and with pGL3-Ank-Luc and pCMV-Renilla reporter. (**D**) Synovial fibroblasts were transfected or not with pcDNA3.1, or pcDNA3.1 construct to overexpress HIF-1 or HIF-2, and with pGL3-HRE-Luc and pCMV-Renilla reporter before being cultured in hypoxia. Results are presented as mean luciferase activity ratio of Firefly/Renilla (±S.D.). Statistically significant differences are indicated as * for *p* < 0.05 from the control in normoxia (**A**,**B**) and as * for *p* < 0.05 from the control in hypoxia (5% O_2_) (**C**,**D**). (*Ank*: inorganic pyrophosphate transport regulator; Cont: control condition meaning transfection with pGL3-Ank-Luc (**A**,**C**) or pGL3-HRE-Luc (**B**,**D**) and pCMV-Renilla reporter; *Hif*: hypoxia-induced factor; HRE: *Hif*-response element; Luc: Firefly luciferase; SD: standard deviation).

**Figure 5 pharmaceuticals-17-00738-f005:**
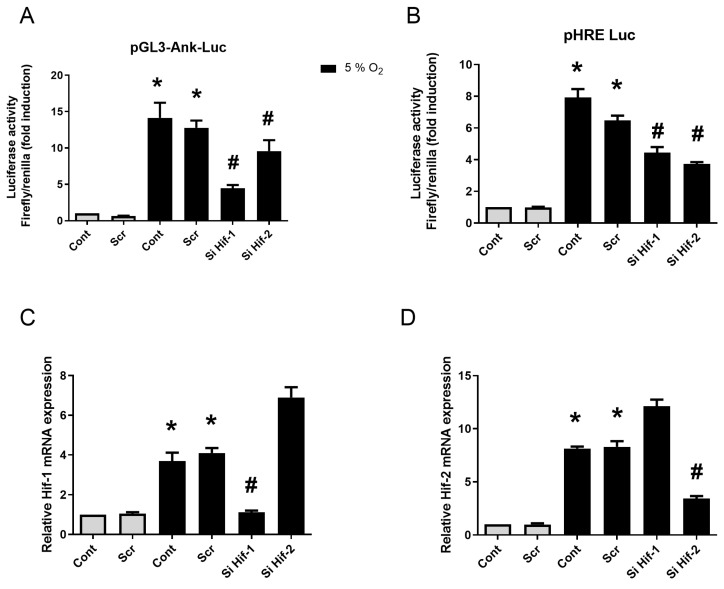
HIF-1, but not HIF-2, increases *Ank* promoter transactivation in vitro. (**A**,**D**) Synovial fibroblasts were transfected or not with siRNA for 12 h, then with plasmids for 24 h, before being challenged with 10 ng/mL of TGF-β1 for 48 h either in normoxia (21% O_2_) or hypoxia (5% O_2_). (**A**,**C**,**D**) Synovial fibroblasts were transfected or not with scramble siRNA, or siRNAs directed against *Hif-1* or *Hif*-2, then with pGL3-Ank-Luc and pCMV-Renilla reporter. (**B**) Control of hypoxia efficiency using synovial fibroblasts transfected or not with scramble siRNA, or siRNAs directed against *Hif-1* or *Hif*-2, then with pGL3-HRE-Luc and pCMV-Renilla reporter. (**A**) Results for *Ank* promoter transactivation are presented in histograms as mean luciferase activity ratio of Firefly/Renilla (±S.D.). (**B**) Results for HRE transactivation are presented in histograms as mean luciferase activity ratio of Firefly/Renilla (±S.D.). Statistically significant differences are indicated as * for *p* < 0.05 from the control in normoxia and as # for *p* < 0.05 from control in hypoxia (5% O_2_). (*Ank*: inorganic pyrophosphate transport regulator; Cont: control condition meaning no siRNA transfection; ePPi: extracellular inorganic pyrophosphate; *Hif*: hypoxia-induced factor; HRE: *Hif*-response element; Luc: Firefly luciferase; Rp29: ribosomal protein 29; RT-qPCR: reverse transcription–quantitative polymerase chain reaction; Scr: scramble RNA meaning non-silencing RNA; SD: standard deviation; Si: means siRNA or small interfering RNA).

**Table 1 pharmaceuticals-17-00738-t001:** Primers used for RT-qPCR.

Gene	Forward Primer	Reverse Primer	GenBank^®^ Accession Number
*Ank*	5′-CAA GAG AGA CAG GGC CAA AG-3′	5′-AAG GCA GCG AGA TAC AGG AA-3′	NM_053714
*Enpp1*	5′-TAT GCC CAA GAA AGG AAT GG-3′	5′-GCA GCT GGT AAG CAC AAT GA-3′	NM_053535
*Hif-1*	5′-AAG TCT AGG GAT GCA GCA CG-3′	5′-GGG GAA GTG GCA ACT GAT GA-3′	NM_024359
*Hif-2*	5′-GCA CCA GCA GTT CAC ACT TG-3′	5′-CTG ACG GTC TTG TCA GGC AT-3′	NM_023090.1
*Rp29*	5′-CTC-TAA-CCG-CCA-CGG-TCT-GA-3′	5′-ACT-AGC-ATG-ATT-GGT-ATC-AC-3′	NM_012876

*Ank*: inorganic pyrophosphate transport regulator; *Enpp1*: ectonucleotide pyrophosphatase/phosphodiesterase 1; *Hif*: hypoxia-induced factor; *Rp29*: ribosomal protein 29; RT-qPCR: reverse transcription–quantitative polymerase chain reaction.

**Table 2 pharmaceuticals-17-00738-t002:** Double-stranded sequences for siRNA silencing.

Name	Sense	Antisense
Scr RNA	5′-CGA UGG GUU CGU GUC GUU U-3′	5′-AAA CGA CAC GAA CCC AUC G-3′
siRNA *Ank*	5′-CUG GCC AAC ACG AAC AAC A-3′	5′-UGU UGU UCG UGU UGG CCA G-3′
siRNA *Enpp1*	5′-GAG GAU GUU UAC UCU AUG A-3′	5′-UCA UAG AGU AAA CAU CCU C-3′
siRNA *Hif-1*	5′-CCC AGC UGU UCA CUA AAG U-3′	5′-ACU UUA GUG AAC AGC UGG G-3′
siRNA *Hif-2*	5′-CGG AGG UCU UCU AUG AAC U-3′	5′-AGU UCA UAG AAG ACC UCC G-3′

*Ank*: inorganic pyrophosphate transport regulator; *Enpp1*: ectonucleotide pyrophosphatase/phosphodiesterase 1; *Hif*: hypoxia-induced factor; Scr: scramble; siRNA: small interfering RNA.

## Data Availability

All data are contained within the article and Appendix A.

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
