# Peer review of "Synovial Membrane Is a Major Producer of Extracellular Inorganic Pyrophosphate in Response to Hypoxia"

_pharmaceuticals, 2024, doi:10.3390/ph17060738_

Round 1
Reviewer 1 Report
Comments and Suggestions for Authors
As the reviewer evaluating this paper, several significant concerns lead me to recommend rejection of the manuscript. Below are the main issues identified:
1. The paper lacks a clear statement of hypothesis or research objectives. While it proposes to investigate the relationship between hypoxia, TGF-β1, and ePPi production in synovial fibroblasts and synovial membrane explants, the specific research questions guiding the study remain unclear. Without a well-defined hypothesis or research objectives, the paper's contributions and significance are difficult to ascertain.
2. Ambiguous Experimental Design: The experimental design lacks clarity, particularly regarding the rationale behind selecting specific experimental conditions and controls. The manuscript fails to adequately justify the choice of TGF-β1 concentration, hypoxic conditions (5% O2), and the duration of exposure in the context of mimicking physiological conditions. Without clear justification, the validity of the experimental setup is questionable.
3. The paper primarily focuses on demonstrating the effects of TGF-β1 and hypoxia on ANK, ENPP1 expression, and ePPi production. However, it lacks comparative analysis with other factors or pathways known to influence ePPi levels. Without such comparative analysis, the specificity and relevance of the findings are limited.
4. Inadequate Molecular Mechanism Exploration: While the paper mentions investigating the involvement of HIF-1 and HIF-2 in hypoxia-induced Ank promoter transactivation, the detailed molecular mechanisms underlying this process are not adequately elucidated. The paper lacks depth in exploring the signaling pathways or downstream effectors involved in mediating the observed effects of hypoxia on ePPi production.
5. The paper presents experimental results demonstrating the effects of TGF-β1 and hypoxia on ANK, ENPP1 expression, and ePPi production. However, the interpretation of these findings is limited, and their significance in the broader context of articular chondrocalcinosis or osteoarthritis pathophysiology is not adequately discussed. Without contextualization within the relevant disease mechanisms, the clinical implications of the findings remain unclear.
6. While the paper addresses an important topic related to the pathogenesis of articular chondrocalcinosis and osteoarthritis, it suffers from conceptual and methodological flaws, as well as a lack of clarity in hypothesis formulation and interpretation of findings. Therefore, I recommend rejecting the manuscript in its current form and suggest that the authors address the aforementioned issues through major revisions before resubmission.
Comments on the Quality of English LanguageNeeds editing.
Author Response
The authors wish to thank the reviewers for their valuable comments and suggestions.
The manuscript has been revised accordingly.
Specific responses to the comments and suggestions can be found below and detailed revisions are in blue in the latest version of the manuscript.
Reviewer #1 #1
- The paper lacks a clear statement of hypothesis or research objectives. While it proposes to investigate the relationship between hypoxia, TGF-β1, and ePPi production in synovial fibroblasts and synovial membrane explants, the specific research questions guiding the study remain unclear. Without a well-defined hypothesis or research objectives, the paper's contributions and significance are difficult to ascertain.
The introduction was revised according to the reviewer’s comment.
- Ambiguous Experimental Design: The experimental design lacks clarity, particularly regarding the rationale behind selecting specific experimental conditions and controls. The manuscript fails to adequately justify the choice of TGF-β1 concentration, hypoxic conditions (5% O2), and the duration of exposure in the context of mimicking physiological conditions. Without clear justification, the validity of the experimental setup is questionable.
Experimental conditions and controls were chosen based on the work published by Cailotto et al. (https://doi.org/10.1186/ar2330) for TGF-β1 and by Zaka et al. (https://doi.org/10.1359/jbmr.090512) for hypoxia.
- The paper primarily focuses on demonstrating the effects of TGF-β1 and hypoxia on ANK, ENPP1 expression, and ePPi production. However, it lacks comparative analysis with other factors or pathways known to influence ePPi levels. Without such comparative analysis, the specificity and relevance of the findings are limited.
TGF-β1 is known to be the best ePPi inducer in articular chondrocytes as shown by Cailotto et al. (https://doi.org/10.1186/ar2330). Here, the novelty of our work is that nothing is currently known about synovial membrane/synovial fibroblasts and ePPi production.
- Inadequate Molecular Mechanism Exploration: While the paper mentions investigating the involvement of HIF-1 and HIF-2 in hypoxia-induced Ank promoter transactivation, the detailed molecular mechanisms underlying this process are not adequately elucidated. The paper lacks depth in exploring the signaling pathways or downstream effectors involved in mediating the observed effects of hypoxia on ePPi production.
We are not sure we understand this comment correctly.
HIF-1 and HIF-2 are the main mediators modulated by hypoxia and trigger the homeostatic response enabling cells to survive and differentiate in low-oxygen conditions (Maes et al. https://doi.org/10.1038/nrrheum.2012.36; Zaka et al. https://doi.org/10.1359/jbmr.090512) and our results clearly showed their parts in the regulation of Ank promoter by hypoxia.
- The paper presents experimental results demonstrating the effects of TGF-β1 and hypoxia on ANK, ENPP1 expression, and ePPi production. However, the interpretation of these findings is limited, and their significance in the broader context of articular chondrocalcinosis or osteoarthritis pathophysiology is not adequately discussed. Without contextualization within the relevant disease mechanisms, the clinical implications of the findings remain unclear.
The manuscript was revised according to the reviewer’s comment. The discussion was improved in order to better describe the possible repercussions.
- While the paper addresses an important topic related to the pathogenesis of articular chondrocalcinosis and osteoarthritis, it suffers from conceptual and methodological flaws, as well as a lack of clarity in hypothesis formulation and interpretation of findings.
We have addressed the reviewer’s comments and modified the paper accordingly.

Reviewer 2 Report
Comments and Suggestions for Authors
The authors of this paper seek to establish the role of hypoxia-induced conditions on the production of calcium pyrophosphate dehydrate (CPPD) crystals in the synovial fluid, particularly in association with articular chondrocalcinosis or osteoarthritis. They investigate the mechanisms underlying CPPD formation, focusing on the regulation of extracellular inorganic pyrophosphate (ePPi) levels by the ANK transporter and ectonucleotide pyrophosphatase/phosphodiesterase 1 (ENPP1). Additionally, the study aims to elucidate the influence of transforming growth factor (TGF)-β1 and hypoxia on ePPi production in synovial fibroblasts and synovial membrane (SM) explants. Through comprehensive experimentation, the authors aim to shed light on the molecular pathways involved and the potential implications for therapeutic interventions targeting inflammatory conditions associated with CPPD crystal depositions. While the paper provides good scientific data, the hypothesis is not clear, and there are some flwas in the manuscript and presentation that prevents the readers to appreciate the quality of the work.
Majors comments :
The crystals formed here appear to follow a different process than those in bone. Could you expand on the genetic regulation of this process of mineralization and what is impaired in cartilage? What is the primary transcription factor involved, and is there a genetic regulatory network? What specific impairments are observed? Are there vesicles involved in crystal formation, and do phosphorus and calcium interact to form the crystals? How does TNAP contribute to crystal formation?
Please provide a clear hypothesis: We hypothesize that hypoxia will increase the production of ePPi in both synovial membrane (SM) and synovial fibroblasts. Elaborate on what is impaired in the pathological process.
The authors utilized one-way ANOVA multiple comparisons followed by Tukey correction or t-test followed by Welch correction. These are parametric tests requiring at least 6 samples per cohort and a normal distribution tested with the Shapiro-Wilk test (for a minimum of 6 samples). Please conduct the distribution test and include the values in supplemental material. Use non-parametric tests accordingly and then rewrite the results and discussion based on the new statistics.
The authors have experimentally shown that the presence of transforming growth factor (TGF)-β1 stimulates the production of ePPi. However, this finding lacks theoretical evidence and should be presented as the conclusion of a clear hypothesis, as it does not align with the introduction.
Minor comments :
For Figure 1: Are there differences observed over time (e.g., between 6 and 12 hours, 12 and 24 hours, etc.) or are they solely related to the control? At what specific time point does Figure 1A represent? The same clarification is needed for Figure 1B (histogram alongside Western blot).
Figure 2 appears comprehensive, yet it's challenging to grasp due to the complexity of the system and the significance it represents. Could you clarify if there are differences in TGF beta levels between hypoxia and normoxia, as well as differences in Ank and ENPP-1? I suggest visually representing only the most important findings in the graph and providing a table in the supplemental material detailing all differences along with the actual values.
Considering Figure 2 as a whole, it's unclear what the main takeaway should be without a clear hypothesis.
Comments on the Quality of English Languagethe English is appropriate
Author Response
The authors wish to thank the reviewers for their valuable comments and suggestions.
The manuscript has been revised accordingly.
Specific responses to the comments and suggestions can be found below and detailed revisions are in blue in the latest version of the manuscript.
Reviewer #2
The authors of this paper seek to establish the role of hypoxia-induced conditions on the production of calcium pyrophosphate dehydrate (CPPD) crystals in the synovial fluid, particularly in association with articular chondrocalcinosis or osteoarthritis. They investigate the mechanisms underlying CPPD formation, focusing on the regulation of extracellular inorganic pyrophosphate (ePPi) levels by the ANK transporter and ectonucleotide pyrophosphatase/phosphodiesterase 1 (ENPP1). Additionally, the study aims to elucidate the influence of transforming growth factor (TGF)-β1 and hypoxia on ePPi production in synovial fibroblasts and synovial membrane (SM) explants. Through comprehensive experimentation, the authors aim to shed light on the molecular pathways involved and the potential implications for therapeutic interventions targeting inflammatory conditions associated with CPPD crystal depositions. While the paper provides good scientific data, the hypothesis is not clear, and there are some flwas in the manuscript and presentation that prevents the readers to appreciate the quality of the work.
Majors comments:
The crystals formed here appear to follow a different process than those in bone. Could you expand on the genetic regulation of this process of mineralization and what is impaired in cartilage? What is the primary transcription factor involved, and is there a genetic regulatory network? What specific impairments are observed? Are there vesicles involved in crystal formation, and do phosphorus and calcium interact to form the crystals? How does TNAP contribute to crystal formation?
We are not sure we understand all the questions correctly. Here, our work is on non-mineralized tissues. If mineralization occurs in these tissues, such as synovial membrane, it is called ectopic mineralization and is of pathological origin. In this work, we showed that HIF-1 under hypoxia and ERK with TGF-β1 were involved in Ank upregulation then ePPi production. Extracellular inorganic phosphate (ePi) crystallizes with calcium to form basic calcium phosphate (BCP) crystal and ePPi crystallizes with calcium to form CPPD crystal. Extracellular vesicles may be involved in crystal formation (Wuthier and Lipscomb https://doi.org/10.2741/3887) because they carry a lot of ANK and ENPP1 and release ePi, ePPi and also crystals in formation, however this topic is not part of our article. TNAP hydrolyzes ePPi to form ePi, then ePi complexes with calcium to form BCP crystals. Excess of ePPi associated with low levels of TNAP promote CPPD crystal formation.
Please provide a clear hypothesis: We hypothesize that hypoxia will increase the production of ePPi in both synovial membrane (SM) and synovial fibroblasts. Elaborate on what is impaired in the pathological process.
The introduction was revised according to the reviewer’s comment.
The authors utilized one-way ANOVA multiple comparisons followed by Tukey correction or t-test followed by Welch correction. These are parametric tests requiring at least 6 samples per cohort and a normal distribution tested with the Shapiro-Wilk test (for a minimum of 6 samples). Please conduct the distribution test and include the values in supplemental material. Use non-parametric tests accordingly and then rewrite the results and discussion based on the new statistics.
This is only a big editorial error from our part, the tests used are non-parametric tests and the statistics shown are made with results are expressed as the mean ± S.D. of at least 3 independent assays. Comparisons were made by analysis of variance, followed by Fisher's t post hoc test, using the GraphPad PrismTM 5.0 software. A value of p < 0.01 was considered significant.
The authors have experimentally shown that the presence of transforming growth factor (TGF)-β1 stimulates the production of ePPi. However, this finding lacks theoretical evidence and should be presented as the conclusion of a clear hypothesis, as it does not align with the introduction.
Our team has previously demonstrated the mechanisms of Ank regulation (mainly through ERK pathway) by TGF-β1 in ePPi production by articular chondrocytes. ePPi has a protective role for cartilage by maintaining the differentiated phenotype of articular chondrocytes (Cailotto et al. https://doi.org/10.1186/ar2330 and https://doi.org/10.1074/jbc.m109.050534). In this work, we clearly showed that hypoxia upregulated ANK thus caused an increase in ePPi production.
Minor comments
For Figure 1: Are there differences observed over time (e.g., between 6 and 12 hours, 12 and 24 hours, etc.) or are they solely related to the control? At what specific time point does Figure 1A represent? The same clarification is needed for Figure 1B (histogram alongside Western blot).
Ank and Enpp1 expressions do not change under control conditions whatever the time point. Here, the time point “zero” is used as control for Figures A and B.
Figure 2 appears comprehensive, yet it's challenging to grasp due to the complexity of the system and the significance it represents. Could you clarify if there are differences in TGF beta levels between hypoxia and normoxia, as well as differences in Ank and ENPP-1? I suggest visually representing only the most important findings in the graph and providing a table in the supplemental material detailing all differences along with the actual values.
We did not measure TGF-β1 production between normoxia and hypoxia.
Considering Figure 2 as a whole, it's unclear what the main takeaway should be without a clear hypothesis.
We wish to let Figure 2 as it is. It allows us to observe:
1/ the effect of normoxia/hypoxia on the Ank/Enpp1 expression and ePPi production
2/ the effect of TGF-β1 with normoxia/hypoxia on the Ank/Enpp1 expression and ePPi production
3/ the respective parts of Ank and Enpp1 with the use of specific siRNAs mainly on ePPi production

Reviewer 3 Report
Comments and Suggestions for Authors
The study paper investigates how hypoxic conditions in the synovial membrane lead to increased production of extracellular inorganic pyrophosphate (ePPi), a crucial factor in the development of calcium pyrophosphate dehydrate (CPPD) crystals seen in osteoarthritis and chondrocalcinosis. Through experiments involving synovial fibroblasts and SM explants, it highlights the role of transforming growth factor (TGF)-β1 and hypoxia-induced factors (HIF), particularly HIF-1, in upregulating ANK and ENPP1, thereby increasing ePPi levels. This mechanistic insight into ePPi production under hypoxic conditions provides a foundation for potential therapeutic targets in conditions related to CPPD crystal formation. The work adds to the realms of rheumatology and pharmaceutical development, shedding light on the pathophysiological mechanisms underlying common yet debilitating conditions like osteoarthritis and CPPD crystal deposition disease.
Minor.
The paper mentions the influence of TGF-β1 and hypoxia on ANK and ENPP1 expression, primarily through HIF-1. I might have missed but discussing other signaling molecules and pathways involved in the hypoxic response, such as NF-kB, MAPK, or PI3K/Akt pathways, could provide a fuller picture of the cellular response to hypoxia.
Comments on the Quality of English Language
English is generally high, reflecting a well-structured and academically sound writing style appropriate for scientific communication. some sentences could be simplified for improved readability without compromising the depth of information.
Author Response
The authors wish to thank the reviewers for their valuable comments and suggestions.
The manuscript has been revised accordingly.
Specific responses to the comments and suggestions can be found below and detailed revisions are in blue in the latest version of the manuscript.
Reviewer #3
The study paper investigates how hypoxic conditions in the synovial membrane lead to increased production of extracellular inorganic pyrophosphate (ePPi), a crucial factor in the development of calcium pyrophosphate dehydrate (CPPD) crystals seen in osteoarthritis and chondrocalcinosis. Through experiments involving synovial fibroblasts and SM explants, it highlights the role of transforming growth factor (TGF)-β1 and hypoxia-induced factors (HIF), particularly HIF-1, in upregulating ANK and ENPP1, thereby increasing ePPi levels. This mechanistic insight into ePPi production under hypoxic conditions provides a foundation for potential therapeutic targets in conditions related to CPPD crystal formation. The work adds to the realms of rheumatology and pharmaceutical development, shedding light on the pathophysiological mechanisms underlying common yet debilitating conditions like osteoarthritis and CPPD crystal deposition disease.
Minor.
The paper mentions the influence of TGF-β1 and hypoxia on ANK and ENPP1 expression, primarily through HIF-1. I might have missed but discussing other signaling molecules and pathways involved in the hypoxic response, such as NF-kB, MAPK, or PI3K/Akt pathways, could provide a fuller picture of the cellular response to hypoxia.
Other pathways modulated by hypoxia and which can also explain its effect on the expression of Ank were added in the discussion (Risbud et al. https://doi.org/10.1097/01.brs.0000186326.82747.13).
Our team has previously demonstrated the mechanisms of Ank regulation (mainly through ERK pathway) by TGF-β1 in ePPi production by chondrocytes (Cailotto et al. https://doi.org/10.1186/ar2330). Hypoxia has also been shown to induce ERK pathway as well as other MAPKs in cartilage (Risbud et al. https://doi.org/10.1097/01.brs.0000186326.82747.13).

Reviewer 4 Report
Comments and Suggestions for Authors
This study evaluated how hypoxia and TGF-β1 influence the production of ePPi by Synovial Membrane and/or synovial fibroblasts. The results bring advancement to the scientific literature, however, the manuscript is not ready to be published. There are some issues that authors need to work on to improve the quality of the manuscript.
1 - It would be interesting if the authors could indicate in the introduction the prevalence of CCA to highlight the clinical significance of the study.
2 - It is important that the authors delve deeper into the causes of the reduced degradation of ePPi.
3 - The authors should reflect on whether it is important how TGF-β1 and Ank influence the production of ePPi.
4 - The justification and hypothesis of the study can be better elaborated
5 - Cell viability data in silencing experiments using siRNA must be included in the text?.
6 - The reports should better discuss the limitations of the study.
7 - The authors should further discuss the signaling pathways involved in regulating the expression of Ank and Enpp1, especially the role of TGF-β1 and hypoxia.
8 - The specific mechanisms by which HIF-1 and HIF-2 regulate Ank expression need to be better addressed in the discussion.
9 - The authors should consider some broader discussion about the therapeutic impact of the study.
10 - The images are adequate, but the graphics could be colored instead of black and white.
Author Response
The authors wish to thank the reviewers for their valuable comments and suggestions.
The manuscript has been revised accordingly.
Specific responses to the comments and suggestions can be found below and detailed revisions are in blue in the latest version of the manuscript.
Reviewer #4
English is generally high, reflecting a well-structured and academically sound writing style appropriate for scientific communication. some sentences could be simplified for improved readability without compromising the depth of information.
This study evaluated how hypoxia and TGF-β1 influence the production of ePPi by Synovial Membrane and/or synovial fibroblasts. The results bring advancement to the scientific literature, however, the manuscript is not ready to be published. There are some issues that authors need to work on to improve the quality of the manuscript.
1 - It would be interesting if the authors could indicate in the introduction the prevalence of CCA to highlight the clinical significance of the study.
The introduction was revised according to the reviewer’s comment.
2 - It is important that the authors delve deeper into the causes of the reduced degradation of ePPi.
The article shows a de novo production of ePPi by the effects of hypoxia and TGF-β1, not a reduced degradation.
3 - The authors should reflect on whether it is important how TGF-β1 and Ank influence the production of ePPi.
The manuscript was revised according to the reviewer’s comment.
4 - The justification and hypothesis of the study can be better elaborated
The introduction was revised according to the reviewer’s comment.
5 - Cell viability data in silencing experiments using siRNA must be included in the text?.
It would be quite cumbersome to introduce these additional experiments into the manuscript, we have chosen not to include these results and to note “data not shown” in the corresponding part.
6 - The reports should better discuss the limitations of the study.
The discussion was revised according to the reviewer’s comment.
7 - The authors should further discuss the signaling pathways involved in regulating the expression of Ank and Enpp1, especially the role of TGF-β1 and hypoxia.
The discussion was revised according to the reviewer’s comment.
8 - The specific mechanisms by which HIF-1 and HIF-2 regulate Ank expression need to be better addressed in the discussion.
The discussion was revised according to the reviewer’s comment.
9 - The authors should consider some broader discussion about the therapeutic impact of the study.
This mechanistic insight into ePPi production under hypoxic conditions provides a foundation for potential therapeutic targets in conditions related to CPPD crystal formation. The work adds to the realms of rheumatology and pharmaceutical development, shedding light on the pathophysiological mechanisms underlying common yet debilitating conditions like osteoarthritis and CPPD crystal deposition disease
The text suggested by Reviewer #4 was added to the section “Conclusions”.
10 - The images are adequate, but the graphics could be colored instead of black and white.
We have chosen to put the graphics in shades of gray because we find that putting colored graphics does not bring a lot of added value but above all entails a cost for those who print and a significant ecological footprint.

Round 2
Reviewer 1 Report
Comments and Suggestions for Authors
This manuscript investigates the role of hypoxia and TGF-β1 in the production of extracellular inorganic pyrophosphate (ePPi) by synovial fibroblasts and synovial membrane explants. While the research addresses a pertinent question in the context of joint diseases, the manuscript is not organized in a conventional manner, making it difficult to follow and understand. Results and discussions are presented immediately after the introduction without a clear explanation of the data and methods used, which is unconventional and problematic for comprehension. Furthermore, the findings do not appear to offer substantial new insights or improvements over existing knowledge in the field.
1. The manuscript does not follow the standard structure of scientific papers. Specifically, the results and discussion sections appear immediately after the introduction, without a dedicated methods section. This unconventional format makes it difficult to understand the study's experimental design, data collection, and analysis processes.
2.The methods used in the study are not clearly described. Key details such as the sample size, specific experimental conditions, statistical analyses, and controls are either missing or inadequately covered. This lack of information hampers the reproducibility of the study.
4.The data is presented in a manner that is difficult to interpret. There is a need for more detailed figures and tables that clearly illustrate the experimental results. Additionally, the statistical significance of the findings should be clearly indicated.
5.The findings do not seem to offer significant new insights into the field. The role of hypoxia and TGF-β1 in ePPi production has been explored in previous studies. The manuscript needs to clearly delineate how this study advances our understanding beyond what is already known.
Major questions and suggested imementations:
1.How were the synovial fibroblasts and SM explants prepared and maintained under normoxic and hypoxic conditions? Provide detailed protocols for cell culture, treatment durations, and conditions.
2.Include a comprehensive methods section detailing the quantitative PCR, Western blot, immunohistochemistry, and RNA silencing techniques used. How were these methods optimized and validated for this study?
4.The study mentions the involvement of HIF-1 and HIF-2 in the transactivation of the Ank promoter. Can you provide more detailed mechanistic insights into how these factors interact with the promoter? Use additional experiments to elucidate these pathways.
Comparative Analysis:
5.Compare the effects of TGF-β1 and hypoxia on ePPi production in more detail. How do these factors interact, and what is the relative contribution of each to the overall increase in ePPi levels?
This manuscript requires significant major revisions to meet the standards of a scientific publication. The organization needs to be revised to include a clear methods section before presenting results and discussion. The experimental design and methods should be described in detail, and the novelty and significance of the findings should be clearly articulated. Addressing these issues will improve the clarity, reproducibility, and impact of the study. Resubmission as a fresh manuscript is recommended after these revisions.
Comments on the Quality of English LanguageNot applicable
Reviewer 4 Report
Comments and Suggestions for Authors
The manuscript underwent important modifications in response to the reviewer’s suggestions. Suggestions that were not accepted were convincingly justified. The manuscript has improved in quality and is suitable for publication. The authors deserve it to be published.